# The Use of Gene Expression Profiling and Biomarkers in Melanoma Diagnosis and Predicting Recurrence: Implications for Surveillance and Treatment

**DOI:** 10.3390/cancers16030583

**Published:** 2024-01-30

**Authors:** James Sun, Kameko M. Karasaki, Jeffrey M. Farma

**Affiliations:** 1Department of Surgical Oncology, Fox Chase Cancer Center, Philadelphia, PA 19002, USA; james.sun@tuhs.temple.edu; 2Department of Surgery, University of Hawaii, Honolulu, HI 96813, USA; kamekok@hawaii.edu

**Keywords:** cutaneous melanoma, gene expression profiling, circulating tumor DNA, personalized medicine, molecular profile

## Abstract

**Simple Summary:**

Cutaneous melanoma is the fifth most commonly diagnosed malignancy in the United States, with a rising incidence. Patients are currently stratified by the eighth edition of AJCC staging using established clinicopathologic features including Breslow depth, ulceration and sentinel lymph node status, among others. Gene expression profile (GEP) tests aim to provide more accurate prognostication based on an individual patient’s own tumor tissue. Several GEP-based tests are currently available and are used in some situations for the clinical management and prognostication of melanomas in the absence of high-quality prospective clinical trials. Randomized clinical trials that study patient outcomes based on the results of GEP tests are not yet available, and it remains unclear how these tests should best be interpreted and utilized in clinical practice. This chapter reviews the GEP tests that are currently available and the supporting literature.

**Abstract:**

Cutaneous melanoma is becoming more prevalent in the United States and has the highest mortality among cutaneous malignancies. The majority of melanomas are diagnosed at an early stage and, as such, survival is generally favorable. However, there remains prognostic uncertainty among subsets of early- and intermediate-stage melanoma patients, some of whom go on to develop advanced disease while others remain disease-free. Melanoma gene expression profiling (GEP) has evolved with the notion to help bridge this gap and identify higher- or lower-risk patients to better tailor treatment and surveillance protocols. These tests seek to prognosticate melanomas independently of established AJCC 8 cancer staging and clinicopathologic features (sex, age, primary tumor location, thickness, ulceration, mitotic rate, lymphovascular invasion, microsatellites, and/or SLNB status). While there is a significant opportunity to improve the accuracy of melanoma prognostication and diagnosis, it is equally important to understand the current landscape of molecular profiling for melanoma treatment. Society guidelines currently do not recommend molecular testing outside of clinical trials for melanoma clinical decision making, citing insufficient high-quality evidence guiding indications for the testing and interpretation of results. The goal of this chapter is to review the available literature for GEP testing for melanoma diagnosis and prognostication and understand their place in current treatment paradigms.

## 1. Introduction

Melanoma is generally a cutaneous malignancy with the highest mortality rate and has demonstrated a rising incidence over the last several decades. Data provided by the National Cancer Institute’s Surveillance, Epidemiology, and End Results (SEER) program estimate that there were 97,610 new melanoma cases in the United States in 2023, a nearly three-fold increase in the rate of new cases per 100,000 people per year since 1975 [1]. It is now the fifth most common newly diagnosed cancer in the United States, representing 5% of all new cancer cases. Fortunately, overall 5-year relative survival remains high at 93.5%, though this severely decreases with regional metastases to 74%.

Improving the treatment of melanoma is only possible through a detailed examination of the molecular and cellular basis of melanoma pathogenesis. A better understanding of the underlying genomic aberrations in melanoma pathogenesis has led to amazing advances in melanoma treatment. By comparing the differences between normal and cancerous cells, the expression of mutated genes can be measured and traced back to the aberrant pathway to identify potential treatment targets. The Food and Drug Administration (FDA) has approved several classes of medications including immune checkpoint inhibitors (ipilimumab, nivolumab, and pembrolizumab), targeted agents against *BRAF* and *MEK* (vemurafenib and cobimetinib, dabrafenib and trametinib, and encorafenib and binimetinib) and most recently *LAG-3* (nivolumab plus relatinib). These therapeutic options for melanoma treatment have changed the treatment landscape for melanoma and improved patient outcomes including overall survival [2].

In addition to identifying targets for treatment, gene expression profiling (GEP) also allows for the identification of groups of genes that when expressed together as a “signature” can serve as a biomarker for the prognosis of certain cancers, including predicting recurrence or metastatic risk [3]. GEP brings us closer to understanding tumors on an individual basis, and to tailoring treatment and surveillance to a specific tumor rather than using generalizations. As such, there are innumerable efforts in this arena to better understand and optimize the use of GEP in clinical decision making. It is important to recognize that currently, neither the American Academy of Dermatology (AAD) nor the National Comprehensive Cancer Network (NCCN) endorse GEP testing [2,4]. The AAD discourages routine GEP testing until better test criteria are defined [4]. Similarly, the NCCN guidelines cite insufficient evidence to include GEP test results in melanoma care [2]. These guidelines further recommend against using GEP testing to guide clinical decision making for stage I melanomas due to the high false-positive rates. A thorough review of the present literature reveals a recurring theme; based on current data available, it is unclear how the results of these tests should be utilized, or even which patients to test. The NCCN allows that prospective trials using GEP tests, especially compared to those using validated pathologic data, would provide a better context for their incorporation into clinical decision making. This chapter will review the available literature on the commercially available GEP tests for melanoma, as well as discuss the controversies over how they should be interpreted.

## 2. Gene Expression Profiling

Mutations at any of the steps in normal cell proliferation pathways can lead to cancer development, and these are termed “driver gene mutations”. A common example is mutations associated with proto-oncogenes (e.g., *Ras*). These genes are transcribed into products associated with normal cellular proliferation; however, gain-of-function mutations or relative over-expression can lead to uncontrolled cellular division [3]. Conversely, tumor suppressor genes have the opposite effect and inhibit cell growth (e.g., *APC* and *TP53*). Loss-of-function mutations or relative under-expressions of these genes also result in uncontrolled cellular division.

Gene expression profiling is a method of analysis that identifies patterns of gene expression in a sample of cells or tissue. In general, the relative expression of genes is measured using mRNA extracted from formalin-fixed paraffin-embedded tumor tissue, then amplified using reverse-transcription polymerase chain reaction (RT-PCR) [3]. With the assistance of artificial intelligence, machine learning or other computational methods, the expression of certain groups of genes termed “signatures” or “profiles” has been demonstrated to help diagnose and prognosticate malignancies [5]. These techniques raise the possibility of bringing cancer care toward personalized treatment, which is tailored to an individual patient based on the specific molecular characteristics of an individual’s tumor. To this end, ongoing research aims to utilize GEP to assist with clinical decisions such as whether to perform a sentinel lymph node biopsy or offer adjuvant immunotherapy. The following sections review the present data for each GEP-based test and discuss the current landscape of GEP for melanoma treatment.

### 2.1. Gene Expression Profiling for Initial Cutaneous Melanoma Diagnosis

The diagnosis of melanoma among all melanocytic lesions can be difficult and nuanced. Annually, more than 4.5 million surgical biopsies are performed on clinically suspicious pigmented lesions based on visual characteristics (i.e., the “ABCDE” criteria of melanoma), of which only a small percentage are pathologically confirmed to be melanomas [6,7]. Furthermore, although histopathologic testing is the gold standard for diagnosis, as many as 15% of lesions remain diagnostically ambiguous with a high degree of discordance between dermatopathologists [8,9,10]. Gene expression profiling has identified gene signatures to help clinicians stratify indeterminate melanocytic lesions for biopsy and differentiate between benign nevi and melanomas. The gene expression profiles discussed below have been validated and demonstrated to improve diagnostic accuracy (Table 1).

### 2.2. Pigmented Lesion Assay

The pigmented lesion assay (PLA, DermTech, La Jolla, CA, USA) is a gene expression profile using two genes: *PRAME* (a preferentially expressed antigen in melanoma) and *LINC* (long intergenic non-coding RNA 518). The PLA is considered positive when one or both genes are identified [11]. Sample cells are collected using an adhesive patch placed over the indeterminate pigmented lesion. The PLA was validated using 389 samples of pigmented lesions, of which 87 were melanomas; the diagnostic sensitivity was 91%, specificity was 69% and negative predictive value was 99% [11]. In the real-world setting, Brouha et al. conducted a study of 3418 suspicious lesions evaluated with the PLA, of which 324 (9.5%) had genetic atypia (*PRAME* or *LINC* expression) [7]. Three hundred and sixteen PLA-positive lesions were biopsied, of which 18.7% were histologically confirmed as melanoma in situ or invasive melanoma, an improvement in the approximately 3–5% diagnostic rate when relying upon visual characteristics to proceed with biopsy [7,12,13]. Furthermore, the same group conducted a study to determine the negative predictive value (NPV). In a cohort of 1781 patients with PLA-negative lesions, 1233 (69%) had a follow-up within 3 years of this test, of which only 10 lesions (0.8%) of these patients were subsequently biopsied and pathologically confirmed to represent melanoma. The NPV was 99.2% (CI 95% = 98.4–99.6%) [14]. In the same study, the authors also enrolled a smaller cohort of 323 patients with PLA-negative lesions for planned repeat PLA testing 2 years after the initial test. Of the 302 lesions that underwent repeat testing, 34 (11%) were PLA-positive and biopsied, of which 3 (1%) were diagnosed as melanoma in situ. Assuming the three melanoma in situ lesions were present at the time of initial diagnosis, the NPV was 99% (95% CI: 97.1–99.8%).

A retrospective study of 472 clinically equivocal pigmented lesions conducted assessments using the PLA test and identified several limitations [15]. Ninety-one biopsies were performed for all PLA-positive and PLA-negative cases with a high clinical suspicion of melanoma. The authors reported discordance between the PLA test and pathologic assessment in 38.5% (35/91) of biopsied cases, leading to unactionable results. Regarding the PLA itself, the study also found that 12.5% (59/472) of specimens “failed” genetic assessment for insufficient genetic material, meaning that none of the three genes were able to be analyzed. Among specimens where one or both of *PRAME* and *LINC* were identified, TERT was not identified in 70.9% (300/472) of these specimens and was only identified in 13 specimens. The authors concluded that the high proportion of unactionable or discordant test results is a barrier to the widespread use of this test, but also question whether or not TERT analysis is of clinical significance.

Recently, the PLAplus test (DermTech, La Jolla, CA, USA) has been under investigation, combining gene expression profiling with gene mutation analysis to improve diagnostic accuracy. PLAplus incorporates telomerase reverse-transcriptase (TERT) promoter DNA driver mutation analysis with the GEP detection of *PRAME* and *LINC* [16]. The authors report that TERT mutations were identified in 70% of PLA-positive lesions diagnosed as melanoma and only 4% of severely dysplastic nevi and non-melanoma lesions. This test increases diagnostic sensitivity to 97%, with a specificity of 62% and an NPV of 99.7% [17]. No further studies of the PLAplus test are available at this time.

### 2.3. MyPath Melanoma

Another diagnostic gene expression profile (MyPath Melanoma, Castle Biosciences, Inc., Friendswood, TX, USA) is a 23-GEP test using quantitative RT-PCR to distinguish benign nevi from malignant melanoma when the histopathologic diagnosis is not clear [18]. The gene signature was identified from a panel of melanocytic lesions, notably excluding metastatic melanomas and non-cutaneous melanocytic lesions. The quantitative expression of 23 genes (including 14 genes involved in melanoma pathogenesis and 9 housekeeping genes) was averaged to develop a numerical score ranging from −16.7 to +11.1, with three defined cutoff values for likely benign, indeterminate and likely malignant lesions. Using a training cohort of 464 samples, the test was able to differentiate benign nevi from malignant melanoma with a sensitivity of 89% and specificity of 93%. Independent validation on a separate cohort of 437 samples reported a sensitivity of 90% and specificity of 91% [18]. MyPath Melanoma was further validated for clinical use in a prospective cohort of 1400 melanocytic lesions, with a reported sensitivity of 91.5% and specificity of 92.5% [19].

There are limitations to this test; importantly, metastatic melanoma was excluded, and gene expression profiles of the primary melanoma can change upon metastasis. Additionally, the volume of melanocytic cells obtained for testing should be considered. Certain melanoma types, such as lentigo maligna or melanoma in situ, are morphologically distributed as single malignant cells or scattered nests of malignant cells, which introduces a higher proportion of benign tissue, potentially leading to false-negative results [19]. Lastly, the 23-GEP test was identified using a large number of common melanoma subtypes, and the authors advocate for further studies to determine the test’s performance in differentiating uncommon nevi and melanoma subtypes [18].

### 2.4. Gene Expression Profiling for Melanoma Prognostication

#### 2.4.1. The 31-GEP Test

The first commercially available prognostic test for melanoma was introduced in 2015 [20]. This test was initially developed to stratify patients into high- and low-risk prognostic subtypes of melanoma independent of pathologic features. This test ultimately became the 31-GEP test known as DecisionDx-Melanoma (Castle BioSciences, Inc., Friendswood, TX, USA). The genes selected for this test were identified from public databases of primary cutaneous or uveal and metastatic melanomas and were all similarly up- or downregulated [20]. The genes were narrowed down by comparing differences in expression between primary and metastatic tumors. Based on the standardized expression of each gene in the signature, melanoma tumors could then be classified based on the risk of metastasis with a score between 0 and 1. The goal of this test is to identify patients at risk of metastasis regardless of the AJCC 8th edition stage. The 31-GEP test categorizes patients into Class 1 or Class 2, representing a low- or high-risk of metastasis, respectively. Subsequently, subclassifications for each group have been introduced, designated as A and B, such that Class 1A (score: 0–0.41) represents the lowest risk, Class 2B (score: 0.59–1.0) represents the highest risk and 1B (score: 0.42–0.49) and 2A (0.50–0.58) represent an intermediate risk of metastasis [20,21,22]. Several studies have been published using archived melanoma specimens [20,21,22,23] or prospectively collected specimens [24,25,26,27,28], and a large meta-analysis [29] has demonstrated consistent the association of the 31-GEP test with melanoma relapse and metastasis. 

Several critiques of the 31-GEP have been raised. One critique is the lack of comparison with known clinicopathologic factors that affect melanoma survival [30,31,32]. Another point is that many clinicians use the test to determine whether or not patients require SLNB, though the test was not designed for this purpose [30]. Vetto et al. demonstrated that a population of patients could be identified using the 31-GEP test with a low (<5%) risk of sentinel lymph node positivity, but it has been noted that the model is solely based on the 31-GEP test score and not adjusted for other factors [30,32,33].

Recently, the 31-GEP test has been integrated with clinicopathologic features (patient age, melanoma mitotic rate, ulceration, Breslow depth, presence of TILs, histologic subtype, and anatomic location (trunk; head and neck; extremity)) that seem to address both weaknesses [34]. A neural network algorithm was employed to determine which parameters had the greatest effect on nodal positivity. The authors reported that the i31-SLNB algorithm provides a more nuanced and personalized assessment of SLN positivity risk than does the use of AJCC 8 T-categories, including the breakdown of T1a tumors into low- and high-risk categories based on clinicopathologic features, potentially addressing another critique of the original 31-GEP test [35].

Going further, a more recent improvement to the 31-GEP test is combination of the i31-SLNB algorithm with a regression model to estimate the personalized risk of recurrence (i31-ROR) [36]. The proposed prognostication algorithm begins the i31-SLNB test, stratifying patients into two categories of <5% and ≥5% risk of SLN positivity. The i31-ROR algorithm is then applied, taking into account the result of the SLNB algorithm, if performed, to estimate 5-year RFS, DMFS and MSS [36]. Both algorithms have been validated using separate cohorts not involved in the design of the models [34,36].

These modifications to the 31-GEP test continue to refine the approach to personalized medicine. It is also important to understand the limitations of the tests, many of which have been discussed in other publications [29,30,32], including the lack of consistency in retrospective versus prospective study design, insufficient follow-up time, disparate patient populations or lack of sentinel node data. We wish to highlight one major limitation, which is that that none of the studies incorporated in creating the models addressed the use of adjuvant systemic therapies, a parallel that must be addressed when comparing them to a similar test, the Oncotype DX^®^ (Exact Sciences, Madison, WI, USA) test used for breast cancer, stratifying patients for adjuvant therapy [37]. The addition of the i31-SLNB algorithm now provides clinicians with a test that can influence treatment decisions. Given the recent changes, it remains to be seen how the 31-GEP test will affect patient management decisions and outcomes.

#### 2.4.2. The Clinicopathologic and Gene Expression Profile Model (CP-GEP, Merlin^TM^)

The CP-GEP model, similarly to the abovementioned i31-SLNB algorithm, was designed with the goal of identifying patients with a <5% risk of sentinel lymph node metastases using known clinicopathologic factors and a different gene signature, and therefore reduces the number of SLNB tests in patients with a low pretest probability of a positive node [38]. The gene signature component of the model was derived from gene expression analysis of a patient cohort containing melanoma patients with nodal metastases [39]. After revision, the gene signature included eight genes (two genes retained from the original study [39]) that play a role in the epithelial-mesenchymal transition process, which has been implicated in facilitating cutaneous melanoma metastases [38,40]. A 754 patient cohort who underwent SLNB was used to create logistic regression models incorporating clinicopathologic features (age, Breslow depth) with this gene signature and is now marketed as the Merlin^TM^ test (SkylineDx, Rotterdam, The Netherlands). The statistical model design is well summarized by Eggermont et al. [41]. The test reports a binary result of low or high risk for nodal metastasis; low risk patients are reported to have <5% risk of metastasis [38]. The authors reported a NPV of 96% for nodal metastasis and area under (AUC) the receiver operating characteristic curve (ROC) of 0.82 (95% CI 0.78–0.86) [38]. The authors also report SLNB reduction rate, defined as the fraction of patients not selected for SLNB using the CP-GEP model, which was 42%. The CP-GEP model was also compared to the Memorial Sloan Kettering Cancer Center (MSKCC) nomogram for predicting SLN metastasis which incorporates 5 clinicopathologic variables (age, Breslow depth, Clark level, biopsy location and tumor ulceration). The MSKCC nomogram had a reported AUC of 0.77 [38].

Two validation studies have been published to assess for risk of nodal metastases. The first studied a cohort of 210 Dutch patients with T1-T4 melanoma, of which 27% had a positive SLN. The authors reported a NPV for SLN metastases of 90.5% (95% CI 77.9–96.2%), with NPV of 100% (95% CI 72.2–100%) in T1, 89.3% (95% CI 72.8–96.3%) in T2 and 75.0% (95% CI 30.1–95.4%) in T3 melanomas. All T4 melanomas were tested as high-risk for SLN metastases [42]. The second study included a cohort of 208 patients from the United States with AJCC 8 T1-T4 melanoma who underwent SLNB [43]. Most patients in this cohort had T1 or T2 melanoma (74%); 21% patients had a positive SLNB. The CP-GEP model correctly identified 40 of 44 patients with positive SLNB as high-risk and 61 of 164 patients with negative SLNB as low risk. For T1-T3 patients, the authors report CP-GEP NPV of 93.8% (95% CI 85.0–98.3) and SLN reduction rate of 33.7% (95% CI 27.1–40.8%). Subgroup analysis of patients age ≥ 65 years yielded similar results.

The CP-GEP model has also been utilized to assess risk for melanoma recurrence. All published studies focus on stage I and II patients with calculation of recurrence free survival (RFS) [41,44,45]. All three studies were performed on cohorts of patients who underwent SLNB and reported significantly worse 5-year RFS among high-risk patients compared to low-risk patients (Table 2).

### 2.5. The 8-GEP Test (MelaGenix)

Another commercially available GEP test is called MelaGenix (Neracare, Frankfurt, Germany) and is only available in Europe. Similar to the above GEP tests, the gene signature was narrowed down to eight genes; however, MelaGenix also utilizes three reference genes [46,48]. MelaGenix assesses the expression of the 11-genes through a qRT-PCR assay to provide prognostic information about melanoma. 

The initial study used a training cohort of 125 patients [46]. They found that a GEP score based on the weighted expression of the included genes generated a continuous score ranging from −0.84 to 3.55. When dichotomized to low-risk (score < 1.3) compared to a high-risk (score ≥ 1.3), there was a statistically significant difference in estimated 5-year MSS. The authors of the study report that it complements AJCC disease stages (ROC AUC = 0.91) to predict MSS. The authors also found that when used as a continuous score (instead of dichotomization), it complemented AJCC staging by providing a range of MSS probabilities (Table 2). Receiver operating characteristic (ROC) analysis resulted in an AUC of 0.91 when the GEP score was combined with AJCC staging for prognostication. Clinical validation in a cohort of 211 patients continued to demonstrate statistically significant association of GEP score with MSS, though the ROC analysis demonstrated less association with prognosis (AUC: GEP score alone 0.65, GEP + AJCC 0.66).

One other study clinically validating the MelaGenix profile was published in 2020 using a cohort of 245 stage II melanoma patients with GEP scores ranging from −0.7 to 3.53, similar to the training study [46,47]. The authors found that among stage II melanoma patients, when the score dichotomized to a low (≤0) and high (>0) GEP score, it is significantly associated with RFS, DMFS and MSS [47]. The 5- and 10-year MSS were 92% in the low GEP score group and 82% and 67% in the high GEP score group, respectively.

Although AJCC staging is the current gold standard for defining prognosis, the MelaGenix clinical validation study posits that with the 10-year MSS difference stated previously, using high-versus low-GEP scores to determine the population most likely to benefit from adjuvant therapy may be more specific than utilizing the classification of AJCC stage IIA/B/C to guide clinical decisions [46]. While the initial stated goal for the MelaGenix assay was to help guide decisions for adjuvant systemic therapy, recent randomized trials such as Keynote 716 have established the benefit of adjuvant immunotherapy for AJCC 8 stage IIB/C melanomas thus changing treatment guidelines [49]. Further studies will be needed to define how this assay can be used.

## 3. Biomarkers for Melanoma Prognostication

### 3.1. Lactose Dehydrogenase (LDH)

At present, serum lactate dehydrogenase (LDH) is the only biomarker associated with melanoma prognostication. Several studies have demonstrated that baseline elevations of LDH greater than twice the upper reference range is associated with worse survival among stage IV melanoma patients, However, only 30–40% of these patients have elevated LDH levels at baseline [50]. Furthermore, elevated LDH levels remain associated with poor outcomes despite advances in treatment options [51,52,53].

A recent study from 2020 investigated whether elevated LDH levels were associated with molecular or immunologic factors that promoted disease progression or treatment resistance [53]. The authors performed a multi-omics analysis of metastatic melanoma patients with known serum LDH levels, which included whole genome sequencing to analyze tumor mutational burden, identify point mutations, copy number variations and promoter methylation. Genomic and metabolomic analyses did not identify significant differences in carbon metabolism, cancer metabolism drivers, glucose metabolism, hypoxia metabolism, KEGG glycolysis, mTOR pathway or choline metabolism pathways. Evaluation of immune pathways also did not demonstrate differences in expression of genes involved in immunologic pathways. The only significant finding was an association between elevated serum LDH and number of metastatic sites. Currently, LDH levels are used for staging advanced melanoma with limited application for treatment monitoring [2].

### 3.2. Circulating Tumor DNA (ct DNA)

Circulating tumor DNA (ctDNA) have been under investigation for the prognostication and monitoring of many cancer types, including melanoma [54]. These are fragmented tumor DNA which are found in patient plasma [55]. The goals of measuring ctDNA, now termed “liquid biopsy”, are for earlier detection of metastatic or recurrent disease and for dynamic monitoring of treatment response and burden of disease by measuring the number of ctDNA fragment copies. Specifically for melanoma patients, low or undetectable ctDNA levels prior to treatment are associated with longer PFS and OS [56]. Several recent studies have also further confirmed its utility for detecting disease recurrence [56,57,58].

#### Signatera^TM^

The Signatera^TM^ assay (Natera Inc., Austin, TX, USA) is a personalized test where whole exome sequencing is performed on a patient’s tumor tissue and peripheral blood to identify up to 16 patient-specific single nucleotide variants (SNV). PCR primers targeting patient-specific SNVs are then used to track ctDNA found in plasma. Full details of the assay are described by Reinart et al. [59]. 

At the time of writing, there are two published studies investigating the Signatera^TM^ assay for melanoma patients. Eroglu et al. studied 69 patients divided into 3 cohorts: (1) stage III patients under observation or receiving adjuvant immunotherapy (n = 30), (2) unresectable stage III/IV patients receiving immunotherapy (n = 29), (3) stage III/IV patients on surveillance after completing planned immunotherapy for metastatic disease (n = 10) [51]. In the first group, patients with detectable ctDNA post-resection (prior to immunotherapy) had significantly shorter distant metastasis-free survival compared to those with undetectable ctDNA (median 4 months vs. not reached, HR 10.77, *p* = 0.01). Recurrent disease was detected on average 3 months prior to identification on imaging. Among the second cohort of patients receiving adjuvant immunotherapy, an increase in ctDNA within 3–11 weeks of treatment was associated with shorter PFS compared to decrease in ctDNA (median 5.7 months vs. not reached, HR 22, *p* = 0.006). All patients in this study with increasing ctDNA developed disease recurrence, whereas all patients with decreasing ctDNA had complete or partial responses. Lastly, among the third cohort of patients under surveillance, all patients with negative ctDNA after treatment remained progression-free over the study period (n = 7, median 14.67 months, range 14.13–18.23 months). The patients with detectable ctDNA during surveillance developed disease recurrence (n = 3); ctDNA was detectable at a median of 3.34 months (range 0.6–6.9 months) prior to clinical detection.

The second study by Brusgaard et al. evaluated 28 stage II and III melanoma patients with the stated goal of evaluating feasibility of detecting ctDNA using the Signatera^TM^ assay in the real-world setting and to assess relationship of ctDNA status with disease stage, treatment and clinical status [60]. The authors reported a strong association of ctDNA detection with disease recurrence as all patients with detectable ctDNA preoperatively (n = 13/28, 46%) or during surveillance (n = 6/20, 30%) went on to develop disease recurrence. 

Overall, the limited studies of the Signatera^TM^ assay are promising for patients under surveillance after resection of high-risk, primary melanoma to detect disease recurrences. The data also suggest utility in checking ctDNA prior to resection to identify those at high-risk for disease recurrence.

## 4. Conclusions and Future Directions

At the time of writing, the latest edition of the NCCN Melanoma Clinical Practice Guidelines recognizes ancillary molecular profiling tests, including the above-mentioned GEP-based platforms, as adjuncts that may facilitate a definitive diagnosis in controversial cases, though emphasizes histologic interpretation by dermatopathologists in addition to utilizing these tests [2]. In terms of prognostication, as mentioned previously, the AAD does not endorse interventions based on results of GEP tests. The NCCN recommends that the commercially available GEP tests marketed to risk-stratify melanomas “do not provide clinically actionable prognostic information when combined or compared with known clinicopathologic factors (sex, age, primary tumor location, thickness, ulceration, mitotic rate, lymphovascular invasion, microsatellites, and/or SLNB status) or multivariable nomograms/risk calculators”. Furthermore, the NCCN Guidelines state that GEP tests are not superior to the abovementioned clinicopathologic factors for prognostication, and that it is still unknown whether these tests reliably predict melanoma outcomes [2].

There remains significant controversy in the role of GEP for melanoma prognostication, which have been articulated in numerous editorials [31,32,61,62,63]. There is consensus that there is considerable opportunity to improve melanoma prognostication, especially among thin melanomas (<0.8 mm) and stage II/III melanomas with the aim of identifying patients in these cohorts with lower risk tumors that may not require invasive procedures or a strict surveillance schedule and conversely, identifying a high-risk cohort that will benefit from aggressive treatment or closer surveillance that would not receive either based on AJCC staging [31,32,61]. Ideally, randomized trials where patients are assigned to treatment groups based on the results of GEP tests are needed to truly understand how these test results should be interpreted. Should these shortcomings be surmounted, there is significant progress to be made in refining melanoma treatment. 

At present, there are no active trials studying GEP tests; however, there are two prospective trials recruiting patients to study the Signatera^TM^ assay (BESPOKE NCT04761783 and PERCIMEL [64] NCT04866680). Both studies aim to evaluate predictive value of the assay on patient outcomes. The BESPOKE trial is studying melanoma, colorectal cancer and non-small cell lung cancer patients receiving immunotherapy to determine how ctDNA levels change with treatment response. The PERCIMEL study will measure ctDNA prior to surgery and will follow ctDNA levels for 2 years after surgery to determine how ctDNA levels correlate to clinical course [64]. The results of these studies will not be available for many years but are eagerly awaited. We anticipate trials of the GEP tests in the future to better define their role in melanoma management.

## Figures and Tables

**Table 1 cancers-16-00583-t001:** Gene expression profiling tests for melanoma diagnosis.

Test	Genes	Statistical Data	Test Modality
2-GEP pigmented lesion assay (PLA)	*PRAME* *LINC00518*	Sensitivity: 91–95% Specificity: 69–91%	Non-invasive skin sample
3-GEP pigmented lesion assay (PLAplus)	*PRAME* *LINC00518* TERT promoter	Sensitivity: 97% NPV: 99.6%	Non-invasive skin sample
23-GEP (MyPath)	Cell Signaling *PRAME*, *S100A7*, *S100A8*, *S100A9*, *S100A12*, *PI3* Tumor Immune Response *CCL5*, *CD38*, *CXCL10*, *CXCL9*, *IRF1*, *LCP2*, *PTPRC*, *SELL* Housekeeping *CLTC*, *MRFAPI*, *PPP2CA*, *PSMA1*, *RPL13A*, *RPL8*, *RPS29*, *SLC25A3*, *TXNLI*	Sensitivity: 90–93.8% Specificity: 91–96.2%	qRT-PCR test of tissue

GEP: gene expression profiling; NPV: negative predictive value; PLA: pigmented lesion assay; qRT-PCR: quantitative reverse-transcriptase polymerase chain reaction.

**Table 2 cancers-16-00583-t002:** Gene Expression Profiling Tests for Melanoma Prognosis.

Test	Studies	Cohort	Results	
31-GEP (Decision-Dx Melanoma^TM^)	Zager 2018 [22]	523 patients Stage I: 50% Stage II: 18% Stage III: 32%	Class 1: 5-y RFS: 88% 5-y DMFS 93%	Class 2: 5-y RFS: 52% 5-y DMFS: 60%
	Greenhaw 2018 [27]	256 patients Stage I: 86% Stage II: 14%	Class 1: 5-y MFS: 93%	Class 2: 5-y MFS: 69%
	Podlipnik 2019 [24]	86 patients Stage IB-IIA: 72% Stage IIB-C: 28%	Class 1: No recurrence: 100%	Class 2: No recurrence: 79%
	Keller 2019 [28]	159 patients Stage I: 60% Stage II: 25% Stage III: 14%	Class 1: 3-y RFS: 97% 3-y DMFS: 99%	Class 2: 3-y RFS: 47% 3-y DMFS: 80%
	Vetto 2019 [33]	838 patients Cohort 1: T1/T2 with SLNB: 326 patients Cohort 2: T1/T2 with SLNB: 512 patients	Cohort 1: Class 1A: 6.2% SLNB+ Cohort 2: Class 1A: 6.3% SLNB+	Class 2B: 8.3% SLNB+ Class 2B: 24.5% SLNB+
	Greenhaw 2020 [29]	1479 patients * Stage IA: 40.3% Stage IB: 17.3% Stage IIA: 10.6% Stage IIB: 7.6% Stage IIC: 2.9% Stage III: 21.1% Unknown: 0.14%	**Stage I** Class 1A 5-y RFS: 97.6% 5-y DMFS: 98.4% Class 1B 5-y RFS: 90.2% 5-y DMFS: 90.0% **Stage II** Class 1A 5-y RFS: 73.0% 5-y DMFS: 89.3% Class 1B 5-y RFS: 83.9% 5-y DMFS: 87.9% **Stage III** Class 1 5-y RFS: 62.9% 5-y DMFS: 72.7%	Class 2A 5-y RFS: 85.0% 5-y DMFS: 90.0% Class 2B 5-y RFS: 76.1% 5-y DMFS: 86.0% Class 2A 5-y RFS: 63.0% 5-y DMFS: 76.5% Class 2B 5-y RFS: 44.3% 5-y DMFS: 60.1% Class 2 5-y RFS: 34.2% 5-y DMFS: 46.1%
i31-GEP	Jarell 2022 [36]	523 patients Validation Cohort Stage IA: 39% Stage IB: 21% Stage IIA: 11% Stage IIB: 8% Stage IIC: 3% Stage III: 18%	Low-risk ** 5-y RFS: 90.5% 5-y DMFS: 94.9% 5-y MSS: 98.0%	High-risk 5-y RFS: 44.7% 5-y DMFS: 52.9% 5-y MSS: 72.6%
8-GEP (MelaGenix)	Brunner 2018 [46]	211 patients Validation Cohort Stage IA: 6.6% Stage IB: 13.7% Stage IIA: 11.8% Stage IIB: 15.2% Stage IIC: 14.2% Stage IIIA: 11.4% Stage IIIB: 15.2% Stage IIIC: 11.8%	Low-risk MSS: 86–98% Intermediate-risk MSS: 40–89% High-risk MSS 47–67%	
	Amaral 2020 [47]	245 patients Stage IIA: 48.2% Stage IIB: 31.8% Stage IIC: 20%	5-y MSS Stage IIA: 94% Stage IIB: 87% Stage IIC: 82% Low score: 92% High score: 82%	10-y MSS Stage IIA: 88% Stage IIB: 82% Stage IIC: 75% Low score: 92% High score: 67%

* Meta-analysis including patients from Gastman 2019 [23], Greenhaw 2018 [24], Hsueh 2017 [25] and a novel 210 patient cohort. ** Test did not stratify according to AJCC disease stages. DMFS: distant metastasis free survival; GEP: gene-expression profiling; MFS: metastasis free survival; MSS: melanoma specific survival; RFS: recurrence free survival; SLNB: sentinel lymph node biopsy.

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
