# Peer review of "The Use of Gene Expression Profiling and Biomarkers in Melanoma Diagnosis and Predicting Recurrence: Implications for Surveillance and Treatment"

_cancers, 2024, doi:10.3390/cancers16030583_

Round 1

Reviewer 1 Report

Comments and Suggestions for Authors

The paper by Sun et al. titled "The Use of Gene Expression Profiling and Biomarkers in Melanoma Diagnosis and Predicting Recurrence, Implications for Surveillance and Treatment" is a review article that discusses the use of gene expression profiling (GEP) for melanoma diagnosis and prognostication. While the review is comprehensive and interesting, I have a few minor concerns that I would like to raise. 

Firstly, the authors mentioned that the American Academy of Dermatology (AAD) and National Comprehensive Cancer Network (NCCN) do not recommend GEP for decision making in melanoma management. It would be helpful if they could better clarify why GEP is not currently recommended and how it might be recommended in the future with any advances. 

Secondly, The information on the LAG-3 antibody could be added in line 53. 

Thirdly, it would be beneficial to add the limitations of PLA. 

Lastly, the numbering of the subtitles appears to be incorrect, with numbers 3 and 4 being swapped.

Author Response

We thank the reviewer for their time and attention to our manuscript. The concerns raised are important to address and we have provided point by point responses below

Firstly, the authors mentioned that the American Academy of Dermatology (AAD) and National Comprehensive Cancer Network (NCCN) do not recommend GEP for decision making in melanoma management. It would be helpful if they could better clarify why GEP is not currently recommended and how it might be recommended in the future with any advances. 

  • We have changed the end of the introduction paragraph as follows (starting from line 65; this replaces the current text)
  • The AAD discourages routine GEP testing until better test criteria are defined.4 Similarly, the NCCN guidelines cite insufficient evidence to include GEP test results in melanoma care. These guidelines farther recommend against using GEP testing to guide clinical decision making for stage I melanomas due to high false-positive rates.2 A thorough review of the present literature reveals a recurring theme; based on current data available, it is unclear how the results of these tests should be utilized, or even which patients to test. The NCCN allows that prospective trials using GEP tests, especially compared to validated pathologic data, would provide better context for their incorporation into clinical decision making. This chapter will review the available literature on the commercially available GEP tests for melanoma, as well as discuss the controversies over how they should be interpreted.

Secondly, The information on the LAG-3 antibody could be added in line 53. 

  • LAG-3/nivolumab plus relatinib have been added to that statement.

Thirdly, it would be beneficial to add the limitations of PLA. 

  • We have added a paragraph discussing PLA testing limitations and a new reference by Ludzik et al as below. We did not find studies evaluating the PLAplus test.
  • A retrospective study of 472 clinically equivocal pigmented lesions assessed with the PLA test and identified several limitations.15 Ninety-one biopsies were performed for all PLA positive and PLA negative cases with high clinical suspicion for melanoma. The authors reported discordance between the PLA test and pathologic assessment in 38.5% (35/91) of biopsied cases, leading to unactionable results. Regarding the PLA test itself, the study also found that 12.5% (59/472) of specimens “failed” genetic assessment for insufficient genetic material, meaning none of the 3 genes were able to be analyzed. Among specimens where one or both of PRAME or LINC were identified, TERT was not identified in 70.9% (300/472) of these specimens and only identified in 13 specimens. The authors concluded that the high proportion of unactionable or discordant test results is a barrier to widespread use of this test, but also question whether TERT analysis is of clinical significance.

Lastly, the numbering of the subtitles appears to be incorrect, with numbers 3 and 4 being swapped.

- Thank you, we will be sure to fix this when reviewing the proof

Reviewer 2 Report

Comments and Suggestions for Authors

Excellent summary of genetic expression profiling for melanoma determination. I would encourage the authors to add a small paragraph on the utility of multi-omics (i.e. coupling genomics, transcriptomics, proteomics, metabolomics) as they mentioned LDH protein levels are a biomarker used for prognostication. I would think an expansion on this point would be warranted.

Author Response

We thank the reviewer for their time in reviewing our manuscript. In response to their suggestion, we have added a paragraph expanding on LDH analyses. We identified a paper from 2020 where the authors performed a multi-omic analysis of metastatic melanoma patients with known serum LDH levels, which is now included as follows:

- A recent study from 2020 investigated whether elevated LDH levels were associated with molecular or immunologic factors that promoted disease progression or treatment resistance.53 The authors performed a multi-omics analysis of metastatic melanoma patients with known serum LDH levels, which included whole genome sequencing to analyze tumor mutational burden, identify point mutations, copy number variations and promoter methylation. Genomic and metabolomic analyses did not identify significant differences in carbon metabolism, cancer metabolism drivers, glucose metabolism, hypoxia metabolism, KEGG glycolysis, mTOR pathway, choline metabolism pathways. Evaluation of immune pathways also did not demonstrate differences in expression of genes involved in immunologic pathways. The only significant finding was an association between elevated serum LDH and number of metastatic sites. 

Reviewer 3 Report

Comments and Suggestions for Authors

This is an important review on the association between gene expression profiling data, which can serve as biomarkers in melanoma diagnosis, prediction of recurrence, and implications for surveillance and treatment of the disease. The manuscript is well-organized and easy to follow. One thing I found lacking is a concluding sentence at the end of the abstract.

Author Response

Thank you for taking the time to read our manuscript and we appreciate your support in our work. We have edited the Abstract and added the following statement to the end: 

  • Society guidelines currently do not recommend molecular testing outside of clinical trials for melanoma clinical decision making, citing insufficient high-quality evidence guiding indications for testing and interpretation of results. The goal of this chapter is to review the available literature for GEP testing for melanoma diagnosis and prognostication and understand their place in current treatment paradigms.